# Assessment of Untargeted Metabolomics by Hydrophilic Interaction Liquid Chromatography−Mass Spectrometry to Define Breast Cancer Liquid Biopsy-Based Biomarkers in Plasma Samples

**DOI:** 10.3390/ijms25105098

**Published:** 2024-05-07

**Authors:** Carmen González Olmedo, Leticia Díaz Beltrán, Verónica Madrid García, José Luis Palacios Ferrer, Alicia Cano Jiménez, Rocío Urbano Cubero, José Pérez del Palacio, Caridad Díaz, Francisca Vicente, Pedro Sánchez Rovira

**Affiliations:** 1Medical Oncology Unit, University Hospital of Jaén, C/Ejército Español 10, 23007 Jaén, Spain; veromg88@gmail.com (V.M.G.); aliciacanojimenez@gmail.com (A.C.J.); rociourbanoc@gmail.com (R.U.C.); oncopsr@yahoo.es (P.S.R.); 2Andalusian Public Foundation for Biosanitary Research in Eastern Andalusia (FIBAO), University Hospital of Jaén, C/Ejército Español 10, 23007 Jaén, Spain; 3Biopathology and Regenerative Medicine Institute (IBIMER), Centre for Biomedical Research (CIBM), University of Granada, 18016 Granada, Spain; jlpalaciosferrer@hotmail.com; 4Fundación MEDINA, Centro de Excelencia en Investigación de Medicamentos Innovadores en Andalucía, Armilla, 18016 Granada, Spain; jose.perezdelpalacio@medinaandalucia.es (J.P.d.P.); caridad.diaz@medinaandalucia.es (C.D.); francisca.vicente@medinaandalucia.es (F.V.)

**Keywords:** early diagnosis, breast cancer, mass spectrometry, hydrophilic interaction liquid chromatography, liquid biopsy, metabolomics

## Abstract

An early diagnosis of cancer is fundamental not only in regard to reducing its mortality rate but also in terms of counteracting the progression of the tumor in the initial stages. Breast cancer (BC) is the most common tumor pathology in women and the second deathliest cancer worldwide, although its survival rate is increasing thanks to improvements in screening programs. However, the most common techniques to detect a breast tumor tend to be time-consuming, unspecific or invasive. Herein, the use of untargeted hydrophilic interaction liquid chromatography−mass spectrometry analysis appears as an analytical technique with potential use for the early detection of biomarkers in liquid biopsies from BC patients. In this research, plasma samples from 134 BC patients were compared with 136 from healthy controls (HC), and multivariate statistical analyses showed a clear separation between four BC phenotypes (LA, LB, HER2, and TN) and the HC group. As a result, we identified two candidate biomarkers that discriminated between the groups under study with a VIP > 1 and an AUC of 0.958. Thus, targeting the specific aberrant metabolic pathways in future studies may allow for better molecular stratification or early detection of the disease.

## 1. Introduction

Breast cancer (BC) ranks as the second most common cause of cancer-related deaths in women overall [1,2,3]. In 2023, BC was the most frequently diagnosed cancer in women, accounting for 31% worldwide [4,5], and the second in Spain for both sexes combined [6] To date, mammography screening is the only effective method for detecting the disease with a high true positive rate, decreasing mortality rates by 41%; nevertheless, it also has disadvantages, such as the cost-expensive resources, exposure to radiation, the breast compression and the final biopsy of the tissue [7,8,9].

It is currently well-known that the behavior of the tumor, prognosis and, therefore, the treatment of BC vary depending on the tumor’s characteristics. Indeed, BC stratification based on the tumor grade, stage and histopathological and molecular characteristics plays a crucial role in determining the rate of survival [10]. Given that histological classification did not reflect the molecular heterogeneity of the disease, the lesions are currently categorized based on standard immunophenotypic analyses considering their mitotic index (ki67), overexpression of estrogen receptors (ER), progesterone receptors (PR), and the human epidermal growth factor receptor 2 (HER2) [11].

Using these molecular markers, BC is mainly classified into three subtypes: hormone receptor (HR)-positive, HER2-positive, and triple-negative breast cancer (TNBC) [11,12,13]. Thus, BC is not a single homogeneous disease but rather a collection of molecularly and clinically diverse subtypes. These subtypes differ in their biological characteristics, clinical behaviors, and responses to treatment. Understanding these subtypes is crucial for tailoring therapeutic strategies, predicting prognosis, and improving patient outcomes. Therefore, there is an urgent need to establish more personalized-based strategies to define non-invasive, cost-effective methods for an early detection of BC molecular subtypes which may ultimately lead to a more accurate prognosis, relapse detection, tailored follow-up and therapy selection.

In this sense, metabolomics offers a powerful and versatile approach for studying the molecular heterogeneity of BC and identifying its distinct subtypes based on their unique metabolic profiles. By unraveling the complex interplay of metabolic pathways and molecular networks associated with different subtypes, metabolomics can contribute to the development of personalized therapeutic strategies and the discovery of novel biomarkers for early detection and monitoring of breast cancer. This molecular pattern could be defined as the metabolomic signature or potential biomarker profile to be detected and analysed in liquid biopsies for early diagnosis of the disease. Several studies have already explored the possibility of using metabolite panels as biomarkers for early diagnosis, tumor characterization and clinical outcome prediction [14,15,16]. Hence, human body fluids such as saliva, urine, serum and plasma have been re-discovered as an excellent source of potential biological markers and, therefore, are analyzed to discover a metabolic profile that may reflect the systemic dysregulation in BC patients’ metabolisms.

The analytical technique chosen for a metabolomic experiment depends both on the sample type and the approach of the study [17,18,19]. Herein, the metabolomic study presented consists of an untargeted liquid chromatography−high resolution mass spectrometry (LC−HRMS) approach, with the final goal being to assess the utility of the hydrophilic interaction chromatography (HILIC) separation technique to find new candidate biomarkers in plasma samples of breast cancer patients. The application of HILIC for bioanalytical LC−HRMS provides us with a more original methodology for the detection of highly polar and hydrophilic substances in biological samples. 

## 2. Results

### 2.1. LC−HRMS Metabolomic Analysis

A non-targeted metabolomic analysis of plasma samples based on a HILIC method coupled to MS in negative electrospray (ESI–) mode is herein exposed. After setting the generic parameter for chromatographic separation and MS detection, we obtained some specific metabolomic fingerprints from the study groups. Figure 1 shows the representative total ion current (TIC) chromatograms of the QCs in comparison with the gradient elution. 

Under the study conditions, we expected to mostly retain polar analytes, as the highly hydrophilic stationary phase column used has been designed to that end (as previously demonstrated [20,21,22]). Moreover, using a high organic content mobile phase (MP) allowed us to remove the highly polar contaminants to clean the column. As can be observed, HILIC TIC chromatograms of the QCs show different elution profiles related to the polarity of the compounds. Specifically, the most intense ions appeared at minutes (min) 1.014, 1.371 and 3.417 (Figure 1a) and decreased from minute 3.4. When comparing the TIC chromatograms with the elution gradient graphic, it seems that the most concentrated or highly expressed compounds eluting at early time points under the conditions of study are of mainly non-polar character, evidenced by the highly organic content of the MP, while polar compounds began to elute in lower concentrations from minute 3, then increased their concentration at 3.3 min. However, the most hydrophilic compounds show very low intensity from minute 4 till the end of the run.

#### Chemometric Analysis

Peak picking and alignment procedures were performed in order to reduce the amount of intensity signals from unrelated ions or unreliable compounds. Based on the data from selected peaks compared in several quality control samples (QCSs), the retention time (R.T) window and mass tolerance were established at 2 s and 10 ppm, and a data matrix featuring 4468 peak intensities was obtained, 1754 of which were monoisotopic ions. After removing features present in the MP, we kept 929 metabolomic signals with a CV < 30% in the QCs, as higher variability within the same analytical replicate would not reflect a reliable behavior within the rest of the samples. To assess the quality of the analytical system performance, we applied the PCA and checked the QC sample clustering. In this case, after the contamination filtering and the removal of features with an unacceptable variability, the close clustering of the QCSs indicated that the natural behavior of the samples and relevant separation between the groups of study are due to biological factors (Figure 2a). Pre-processing approaches by QC normalization of the data matrix, log transformation and pareto scaling were chosen as the best options to obtain a normally distributed mode. The quality of the model to discriminate between the groups of study was determined using R^2^ and Q^2^ from the PLS-DA. In this regard, models which display R^2^ ≥ 0.7 and Q^2^ ≥ 0.4 with variance between these parameters < 0.3 have been previously reported as featuring efficient diagnostic power [23]. In our case, the combination of significant candidate metabolites showed high values for the goodness of prediction (R^2^) and the predictability of the model (Q^2^), while the PLS-DA illustrated a good discrimination between the BC and HC samples (Figure 2b and Table 1). Further analyses in larger cohorts might explain the metabolome differences between the four BC molecular subtypes according to the TNM staging system, as suggested by our data in Appendix A.

### 2.2. Selection of Potential Biomarkers

Potential biomarkers were detected within the groups using univariate and multivariate analyses. First selection criteria by *t*-test (*p* corrected value by false discovery rate [FDR] < 0.05) and fold change (> 2) allowed for the detection of 347 differentially expressed metabolic features between the BC and HC samples. Second selection criteria, based on the VIP values > 1, estimated the importance of each selected variable in the PLS-DA model projection. As a result, 27 out of the 77 molecular signals that met the conditions in the whole BC cohort were selected for biomarker evaluation, model creation and identification (Table 2 and Appendix A). 

#### 2.2.1. Identification of Potential Biomarkers

According to the data collected during the chromatography and MS analysis (accurate mass, R.T and MS/MS patterns), we could achieve a tentative identification with levels of 2 and 3 (according to the Shymansky classification [24]) of two candidate metabolites (Table 3). The molecular formula per candidate was provided by comparison of the experimental fragment interpretation against several spectral data bases, as mentioned in methodology.

#### 2.2.2. Biomarkers’ Evaluation

ROC curves were used to evaluate the classification and diagnostic power of the candidate metabolites. In this case, univariate ROC curves per each candidate were performed in order to assess their potential clinical utility in terms of AUC; the AUC values obtained ranged from 0.7 to 0.9. Considering that AUC values greater than 0.75 indicate a feasible predictor model, the identified metabolites of Table 3 were selected for model creation by further multivariate analyses. 

#### 2.2.3. Model Creation

Due to the multifactorial character of cancer, a multivariate model may improve levels of discrimination and confidence by combining multiple individual potential biomarkers. In this regard, we applied UVA and MVA criteria to combine the two identified candidate biomarkers, and we obtained an AUC of 0.958 (95% CI: 0.927–0.987) for BC diagnostic capability (Figure 3).

## 3. Discussion

Lately, the utilization of mass spectrometry-based techniques to detect metabolic profiles in blood has emerged as a selective and sensitive tool to improve diagnosis of malignant diseases, including several types of cancers (such as colorectal, gastroenterological or pancreatic cancers) [25,26,27,28]. Specifically, metabolomic analyses of BC have widely demonstrated that metabolic dysregulation in cancer might be detected in a simple and cost-effective manner which may yield potential biomarkers of the disease behavior after further validation. As an example, untargeted metabolomics have previously shown that altered molecular pathways related to BC initiation involve the biosynthesis of unsaturated fatty acids, aminoacyl-tRNA biosynthesis, carnitine metabolisms, phenylalanine, tyrosine and tryptophan biosynthesis, as well as nicotinate and nicotinamide metabolisms [16,29,30]. Moreover, this strategy could also be used in the search for predictive and prognostic biomarkers when combined with temporal statistical methods that may reinforce its high value as a tool for deciphering cancer behavior [14]. However, targeted metabolomic strategies conducted to determine the diagnostic performance of the huge amount of metabolomic data found by these approaches are scarce, using quite small sample sizes for the validation sets [31,32].

In this work, we addressed the discovery of new potential biomarkers in liquid biopsies of breast cancer patients by using an untargeted HILIC-HRMS-based metabolomic and negative ionization. Our main goal is to assess this strategy for that purpose; thus, under the study conditions, we could observe that most of the significantly altered compounds between the BC and HC groups eluted from minute 3 to minute 4, which would correspond to those analytes with intermediate polarity. In contrast, compounds that eluted at early times, covering a quite short R.T. range (from minute 1 to minute 2), would correspond to non-polar analytes, as evidenced by the percentage content of the MP at that time range [21,22]. In this sense, when analyzing TIC chromatograms of the QCSs under study, clear differences appeared in the elution range according to the compounds’ polarity characteristics. To discriminate the metabolites that significantly differed between the HC and BC patients, UVA and MVA were performed. After a series of data pre-processing approaches, the model obtained displayed an efficient diagnostics power with low variance between the R^2^ and Q^2^ from the PLS-DA, as well as high individual and multivariate values of the AUC. Specifically, a great number of polar compounds eluting from minute 3 to minute 3.5 were found to have differential expressions between the study groups, although, at first sight, the polar TIC intensities did not seem to be as high as the non-polar compounds. This result would suggest that non-polar compounds were highly expressed in plasma samples, while polar compounds appeared with lower intensity. In this regard, it should be noted that the following limitations were found when using the HILIC ESI—mode coupled to HRMS: (1) the detection of non-polar components near to the void volume; (2) that most of the significantly altered mass signals were obtained at similar retention times, so it could be possible that a pattern ion fragmented at the ionization source, producing the rest of the fragment ions. 

Nevertheless, the significance of a metabolite as a biomarker is not directly related with its intensity or concentration in a sample. Low concentrated metabolites with low TIC intensities could be of significant biological importance, as their expressions differ enough between the groups of study, i.e., the HC and the BC patients. 

Concerning the identification of specific potential biomarkers, the molecular properties based on accurate mass, the MS/MS spectra and R.T allowed us to define a tentative empirical formula for two candidate metabolites (*m*/*z*: 948.2027, R.T: 3.2 min and *m*/*z*: 914.2331, R.T: 4 min). As previously reported, by using the formic acid content in the MP, we observed the occurrence of intense [M-H] and [M+HCOO]− ions under negative ESI and acidic conditions [33]. Despite the limited evidence for direct implications in cancer development of the long fatty acids (3-isopropenylpimeloyl-CoA and the 2,6-Dimethylheptanoyl-CoA) and the carboxylic acid (6-{[2-(4-{[3-({3,4-dihydroxy-4-[(1H-indole-3-carbonyloxy)methyl]oxolan-2-yl}oxy)-4,5-dihydroxy-6-(hydroxymethyl)oxan-2-yl]oxy}phenyl)-4-oxo-3,4-dihydro-2H-1-benzopyran-7-yl]oxy}-3,4,5-) identified in our work, it is well-established that alteration of cellular metabolism is a hallmark of cancer [34]. In this sense, previous research has shown that aberrant lipid metabolism in cancer could potentially affect processes such as the tumorigenesis or immune evasion of breast cancer [35]. In fact, certain derivates of the coenzyme A have been implicated in processes relevant to cancer development, while our tentatively identified candidates, the 3-isopropenylpimeloyl-CoA and the 2,6-Dimethylheptanoyl-CoA, might be involved in metabolic pathways related to cancer metabolism, such as the biotin biosynthesis or the beta-oxidation of branched-chain fatty acids [36,37,38,39]. Furthermore, regarding the carboxylic acid identified in our study, although no direct implication of this compound with BC has been reported yet, it has been shown that hydroxycarboxylic acid receptors are important for controlling the balance of lipid/fatty acid metabolism in breast cancer cells [40]. Therefore, targeting the specific metabolic or signalling pathways involving these identified molecules and validating the molecular alterations in a bigger cohort of BC patients would be critical to deciphering the direct role of these novel candidate metabolites in BC initiation.

## 4. Materials and Methods

### 4.1. Sample Collection and Preparation

Plasma samples of two experimental groups were analysed in this research: 136 samples from healthy controls (HC) and 134 samples from breast cancer patients. The HC group consisted of Caucasian subjects consecutively recruited from the general community in the same period of time with a body mass index (BMI) in the normal to overweight range aged 18−63 years old. The BC patients were also Caucasians; the majority of them were overweight or obese with an age range between 25 and 84 years old (Table 4). The sample collection from these patients was carried out before starting any therapeutic treatment. Samples were obtained at the University Hospital of Jaén with the written informed consent of all the participants and the approval of the ethics committee.

After blood collection by usual venipuncture, samples were centrifuged for 15 min at 3000 rpm and 4 °C. The supernatant was carefully aspirated and stored at −80 °C until the LC−HRMS analysis.

### 4.2. Metabolomic Analysis

#### 4.2.1. Metabolite Extraction

All plasma samples were kept at 4 °C throughout the analytical process. Proteins were removed from the samples to avoid ion suppression in the analytes of interest. Protein precipitation was achieved using methanol 1:5 (plasma/MeOH) and shaking for 60 s. Samples then required centrifugation at 4 °C and 13,300 rpm for 15 min; the supernatants were then lyophilized (Savant, Holbrook, NY, USA), and the dry residues were reconstituted in 50% water/acetonitrile (AcN). These solutions were transferred to the analytical vials, stored in the autosampler at 4 °C, and analyzed by LC−HRMS.

#### 4.2.2. LC−HRMS Conditions

Chromatographic separation was performed by the Agilent Series 1290 LC system (Agilent Technologies, Santa Clara, CA, USA) using a Waters XBridge BEH Amide column (2.1 mm × 150 mm; Waters Corporation, Milford, MA, USA) kept at 35 °C. Mass detection was achieved using an AB SCIEX Triple TOF 5600 quadrupole-time-of-flight spectrometer (QTOF-MS) in ESI– mode (AB SCIEX, Concord, ON, USA).

The injected sample volume was 3 μL. The MP consisted of 0.1% formic acid—90:10 water/AcN (eluent A) and 0.1% formic acid—90:10 AcN/water (eluent B). The gradient elution was performed as follows: 0–0.1 min 99% eluent B, 0.1–7 min 99% eluent B, 7–7.10 min 30% eluent B, and 7.10–10 min 99% eluent B. The elution flow rate was 0.4 mL/min.

The in-batch sequence of the samples was established based on random number generation to avoid any possible time dependent changes in HPLC-MS chromatographic profiling. Blank solvents based on the MP samples and QCS were also analyzed during the run. The QCSs were obtained by mixing small aliquots of all the biological samples under study and systematically injected every 10 samples for further evaluation of the stability, quality and integrity of the system [41]. The analysis of the MP influence on the metabolomic profile of each group allows the detection of contamination either from the solvent impurities or the extraction procedure [19].

#### 4.2.3. Data Set Creation

Peak picking and alignment procedures precede the data set creation in order to diminish the non-linear shifts in both the R.T and *m*/*z* of the LC−HRMS chromatograms [42]. To this end, the R.T and *m*/*z* variability of the raw data was evaluated within the QCSs by PeakView (PV) software (version 1.0 with Formula Finder plugin version 1.0, AB-SCIEX, Concord, ON, USA). 

MarkerView software (version 1.2.1, AB SCIEX, Concord, ON, USA) was used for processing the LC−HRMS raw data. MarkerView (MV) allowed us to perform peak extraction and alignment, as well as data filtering, in order to generate a data matrix with the *m*/*z*, R.T and integrated intensity that determines each particular ion. The MV data extraction parameters are as follows: (1) R.T length in time (minutes or min) refers to the range of time for the peak extraction in which the analytes were eluted from the separation column; (2) R.T and mass window are the minimum values detected in the RT range that define a single peak; (3) R.T and *m*/*z* tolerances are the values by which a particular peak is considered as the same feature in the peak alignment. In this study, the data extraction parameters were set as follows: (1) R.T range 0.8–9 min, (2) R.T and mass window of two scans and 0.02 Da for the peak extraction and (3) R.T and *m*/*z* tolerances of 0.2 min and 10 ppm for the peak alignment. Moreover, intensity values lower than 70 cps were established as background noise.

The identification of true molecular features is based on the accuracy of mass measurement to group ions related to the charge-state envelope and isotopic distribution. The minimum sample size of the groups of study was considered to filter the extraction to only masses that appeared at least in 14 samples. The data matrix obtained was then processed.

#### 4.2.4. Data Pre-Treatment

From the data matrix, only the extracted peaks that represented monoisotopic ions were selected for the data pre-processing to enhance the identification of true biological features and to reduce mass redundancy. Detection of differentially expressed mass signals between the MP and study groups (HC and BC) allowed for filtering out the influence of non-specific compounds or contaminants by a FC > 2 and *p*-value < 0.05, preserving the true biological mass signals of the data matrix. Data pre-treatment comprises data normalization, filtering, transforming and scaling [43]. These steps were carried out using Metaboanalyst 6.0 (last version of the Web Server Software, Canada, RRID: SCR_015539) [44]. In order to obtain a more Gaussian-type distribution of the data matrix, different approaches for data normalization, scaling and transformation were assessed.

#### 4.2.5. Analytical Validation and Outliers’ Detection

Principal components plot and partial least squares analyses were performed to observe the analytical system behavior and to detect outlier samples. Moreover, the standard measure of relative variance or relative standard deviation (RSD) was used to detect high variance in features within the QCSs. For untargeted analysis in biomarker discovery, the Food and Drug Administration (FDA) proposed RSD > 30% as being sufficiently high enough to warrant removal of those variables with unacceptable reproducibility. Model diagnostics were determined by how well the model fit the data (R^2^ or goodness of fit) and how good the predictive ability of the model (Q^2^ or goodness of prediction) proved to be. Thus, cross-validation simulates the true predictive power of the model [45].

#### 4.2.6. Data Treatment

The detection of differential metabolites between the groups of study from the data set was processed by using the “Statistical Analysis” module of Metaboanalyst. Candidate biomarkers were selected based on the following three principal UVA and MVA criteria: Student *t*-test (*p* corrected value by FDR < 0.05), fold change (FC > 2) and VIP scores according to the rule greater than one (VIP > 1). 

#### 4.2.7. Biomarkers’ Evaluation and Model Creation

Last, we assessed the predictive capability of the candidate metabolites as biomarkers by the univariate AUC-ROC as well as in combination in a multivariate model [23]. To that purpose, we used the module “Biomarker Analysis” in Metaboanalyst with 100 cross-validations using a balanced Monte-Carlo sub-sampling approach and the PLS-DA algorithm.

#### 4.2.8. Molecular Identification

The PV software (version 1.0 with Formula Finder plug-in version 1.0, AB SCIEX, Concord, ON, Canada) was used to evaluate the LC−HRMS data obtained in LC–QTOF-MS and to estimate the elemental formulae of pre-selected candidates based on the parent ion mass, isotopic profile of the parent ion and the MS/MS mass spectra. A further search for structural identification was achieved by comparison of the experimental fragmentation with that provided in spectral databases (SIRIUS Software 5.8.6, CEU mass mediator, NIST 2014, MS/MS, MassBank, Metlin, Human Metabolome database).

## 5. Conclusions

The detection of 347 metabolomic features that discriminate HC and BC patients was possible using our HILIC-HRMS strategy. Further work would be necessary in order to set up the identification of more significant mass signals and the strategy presented here. Nevertheless, according to our criteria, and despite the limitations found, the untargeted LC−HRMS has been suitable in regard to detecting a metabolomic profile that successfully discriminates between the HC and BC patients in a cost-effective and minimally invasive way. Therefore, improvements to different LC−HRMS-based metabolomic approaches to define a metabolomic signature related to the presence of the disease would be conclusive in enhancing the early diagnosis of breast cancer and its molecular stratification, or to predict the relapse after a therapy cure, hopefully leading, in the near future, towards better and more precise treatment for this pandemic disease.

## Figures and Tables

**Figure 1 ijms-25-05098-f001:**
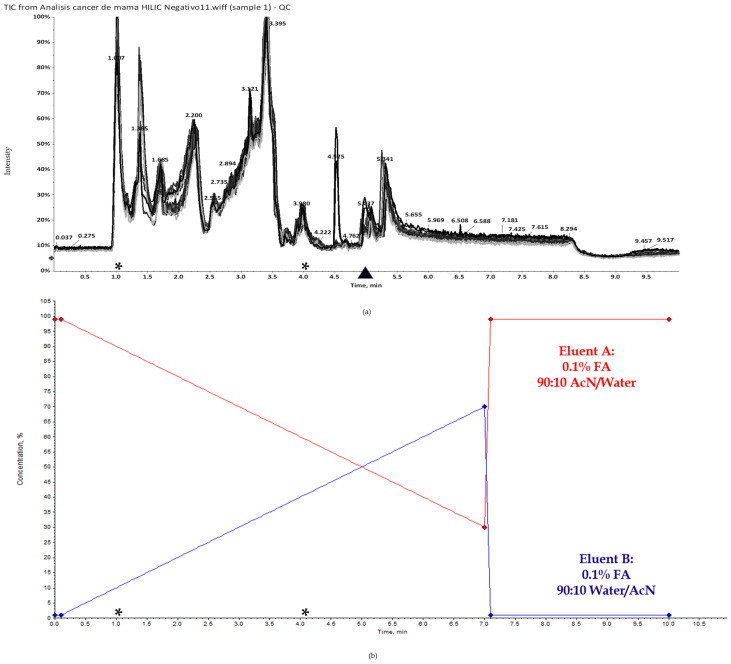
Representative LC−HRMS TICs of pooled QC samples (**a**) in comparison with the gradient elution graphic (**b**). HILIC TIC chromatograms showed different peak profiles along the run which illustrate the elution of the compounds according to their polarity characteristics. * The retention time range where the significantly altered metabolites between breast cancer patients and healthy controls were found to elute *.

**Figure 2 ijms-25-05098-f002:**
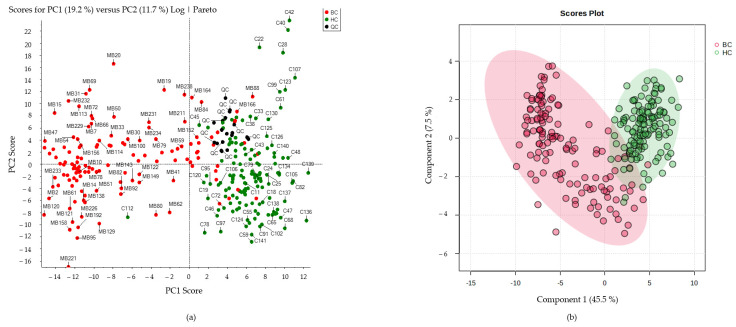
(**a**) The clustering of the QC samples in the middle of both sets of samples in the PCA shows a good natural separation between the groups under study: breast cancer (BC) and healthy controls (HC). The PLS-DA score plots based on the LC−HRMS of plasma samples from BC suggest metabolome differences in comparison with the HC group (**b**).

**Figure 3 ijms-25-05098-f003:**
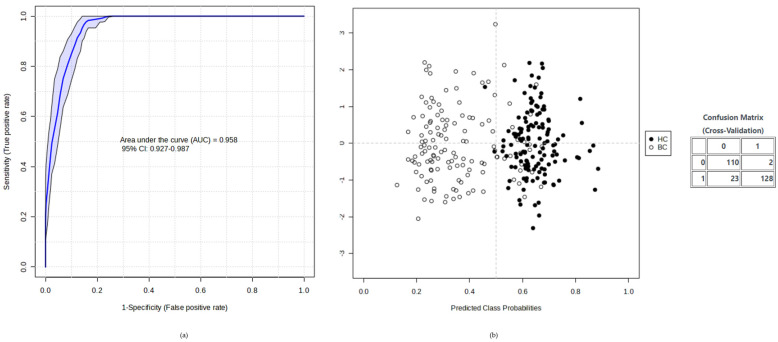
(**a**) Multivariate ROC curve plots from the average of 100 cross-validations for the model combination of two tentatively identified candidate metabolites. (**b**) Classification using the average of predicted probabilities of each group of samples provided a confusion matrix where 110 BC patients were correctly classified and 23 were misclassified; 128 HC were correctly classified and two were misclassified.

**Table 1 ijms-25-05098-t001:** Statistical validation of the PLS-DA models based on the comparison of breast cancer (BC) to healthy controls (HC).

PLS-DA Comparison	Explained Variation (%)	# Components	Accuracy	R^2^	Q^2^
BC-HC	64.3	5	0.973	0.890	0.842

BC: breast cancer patients; HC: healthy controls; #: number; R^2^ and Q^2^ parameters showed that no over-fitting was observed, and these models are acknowledged for successful discernment between HC and BC patients.

**Table 2 ijms-25-05098-t002:** List of candidate metabolites selected for further identification and their respective *p*-value, fold change, VIP score and AUC.

*m*/*z*	R.T (min)	p-Value (FDR)	Fold Change (BC/HC)	VIP	AUC
391.1514	1.03	2.24 × 10^−59^	0.06	1.67	0.97
970.1298	3.19	1.29 × 10^−49^	0.13	2.05	0.93
1012.111	3.16	8.67 × 10^−56^	0.14	2.41	0.94
948.2027	3.16	3.53 × 10^−49^	0.15	2.07	0.93
529.0881	2.82	3.73 × 10^−22^	0.17	1.15	0.83
780.9692	2.82	9.43 × 10^−23^	0.18	1.14	0.84
445.1282	2.81	2.16 × 10^−22^	0.19	1.19	0.83
948.8976	2.82	4.53 × 10^−23^	0.19	1.14	0.84
958.3061	3.29	6.52 × 10^−42^	0.2	1.79	0.91
890.3257	3.26	1.82 × 10^−39^	0.22	1.91	0.91
818.3999	3.15	3.25 × 10^−34^	0.26	2.13	0.90
822.3952	3.14	8.79 × 10^−31^	0.27	2.1	0.89
744.4759	3.14	6.02 × 10^−30^	0.29	2.26	0.90
508.8292	2.56	1.93 × 10^−20^	0.31	1.31	0.85
431.1842	1.09	6.84 × 10^−16^	2.15	1.16	0.75
448.1705	1.08	2.99 × 10^−16^	2.64	1.06	0.77
914.2331	3.96	1.71 × 10^−18^	5.58	1.2	0.86
969.2408	3.98	7.03 × 10^−18^	6.26	1.1	0.87
674.7234	3.23	1.74 × 10^−19^	6.59	1.01	0.84
395.0961	4	3.86 × 10^−22^	6.92	1.05	0.88
303.923	3.9	1.92 × 10^−22^	8.44	1.25	0.86
674.726	3.51	8.03 × 10^−39^	15.02	1.01	0.92
684.7542	3.21	6.63 × 10^−42^	28.11	1.37	0.93
754.7421	3.34	1.03 × 10^−48^	88.94	1.62	0.93
516.8156	3.14	7.53 × 10^−37^	112.86	1.37	0.92
752.7279	3.14	3.28 × 10^−48^	123.83	1.58	0.94
500.8534	3.24	4.12 × 10^−45^	246.87	1.7	0.94

*m*/*z*: mass/charge ratio; R.T: retention time; FC: fold change > 2 indicates that the average normalized peak area ratio in breast cancer (BC) samples is larger than that in healthy controls (HC); FC < 0.5 indicates that the average normalized peak area ratio in HC is larger than that in BC samples; VIP: variable of importance in projection; AUC: area under the receiver-operating characteristic curve.

**Table 3 ijms-25-05098-t003:** Differential tentative identified metabolites between breast cancer (BC) patients and healthy controls (HC).

*m*/*z*	R.T (min)	Molecular Formulae	Adduct	Tentative ID	Mass Error (ppm)
948.2027		C31H50N7O19P3S	[M-H]	3-isopropenylpimeloyl-CoA	1.6
	3.2	C30H48N7O17P3S	[M+HCOO]−	2,6-Dimethylheptanoyl-CoA	
914.2331	4.0	C41H43NO20	[M+HCOO]−	6-{[2-(4-{[3-({3,4-dihydroxy-4-[(1H-indole-3-carbonyloxy)methyl]oxolan-2-yl}oxy)-4,5-dihydroxy-6-(hydroxymethyl)oxan-2-yl]oxy}phenyl)-4-oxo-3,4-dihydro-2H-1-benzopyran-7-yl]oxy}-3,4,5-trihydroxyoxane-2-carboxylic acid	3

*m*/*z*: mass/charge ratio; R.T: retention time; ID: identification.

**Table 4 ijms-25-05098-t004:** Characteristics of the two groups of study: breast cancer patients (BC) and healthy controls (HC).

Study Group	BC	HC
*n*	134	136
Age (y.o)		
Median	49 (25–84)	46 (18–63)
Mean	52.05	43.75
S.D	11.62	11.2
BMI (kg/m^2^)		
Mean	26.84	24.40
S.D	5.54	2.67
Medication		
Yes	68	6
No	66	130
Stage		
IA	2	
IIA	54	
IIB	48	
IIIA	15	
IIIB	6	
IV	4	
Not Available	5	
Phenotype		
LA	21	
LB	64	
HER2	35	
TN	14	
Climacteric		
Premenopause	69	
Perimenopause	60	
Postmenopause	5	

BC: breast cancer patients; HC: healthy controls; *n*: sample size; y.o: years old; S.D: standard deviation; BMI: body mass index; LA: luminal A; LB: luminal B; HER2: human epidermal growth factor 2 positive; TN: triple negative.

## Data Availability

All data generated or analyzed during this study are included in this published article and its Appendix A. Raw data are not publicly available due to ethical restrictions, since they contain information that could compromise the privacy of research participants, but they are available from the corresponding author on reasonable request.

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
