# Peer review of "Assessment of Untargeted Metabolomics by Hydrophilic Interaction Liquid Chromatography−Mass Spectrometry to Define Breast Cancer Liquid Biopsy-Based Biomarkers in Plasma Samples"

_ijms, 2024, doi:10.3390/ijms25105098_

Round 1

Reviewer 1 Report

Comments and Suggestions for Authors

The manuscript is devoted to untargeted plasma metabolic profiling of patients with breast cancer. In general manuscript is interesting and provides some new metabolic features on plasma of breast cancer patients. However, some questions should be clarified to make the manuscript more understandable.

1) The title and abstract (line 27) contain misleading information about “Liquid Biopsy”, however, when reading Materials and Methods we see only plasma as biological sample (lines 252-253). Please, change this mistake and use Plasma both in the title and abstract.

2) Lines 21-22 contain information that “Breast cancer (BC) is the first cause of death by tumour pathology in women worldwide”, then lines 38-39 contain some other information “Breast cancer (BC) ranks as the second most common cause of cancer-related deaths in women overall”. So, the first or the second?

3) Figure 1. There are several questions to this figure. Are these chromatograms the sum of all healthy controls vs the sum of all breast cancer patients? Presumably not. If so, how did the authors decide which of the chromatograms was the most representative? In general, these chromatograms do not carry any valuable semantic meaning. If the authors want to discuss the retention time of some chemical groups of compounds, they should replace this figure into Suppl. and discuss some features in general, or add some illustrative information on the figure to show where the compounds of interest were eluted.

4) Figure 2. My opinion is that Fig.2 b is the most important figure here, because it is clear and answer the question how well groups of patients and controls separate from each other. The meanings of other figures are unclear. Fig 2 a requires some more information than just lines 117-118. Figures 2 c-f are completely unclear. Figure 2 b has already answered the question on clustering patients from controls. The next interesting feature could be related with how four groups of patients could be separated from each other but not from controls. So, figures 2 c-f are unnecessary.

5) If the authors decided to conduct some specific clustering on four groups of patients according to molecular subtypes it would also be interesting to analyze if the same clustering could be performed according to the stage of cancer since groups with stages IIA, IIB and IIIA are relatively similar in amount and could be statistically analyzed. The subsequent analysis of the biomarker profiles which were different in various groups seems very interesting and could significantly improve the manuscript.

6) Lines 111-115 contain information about QC samples. When searching for some detailed information in Material and methods section, the following information was found (lines 290-292): “Blank solvents based on the MP samples and QCs were also analyzed during the run. The QCs were obtained by mixing small aliquots of all the biological samples under study. QCs evaluation provides with a measurement of the stability, quality and integrity of the system.” The mentioned approach seems to be unusual and the explanation of its use is important. Or some relevant references should be added, since [19] contain some general aspects. Were any internal standards added to the samples to control the analysis?

7) Were the absolute peak areas used in the statistical analysis? No normalization procedure on any standard?

8) Section 2.2.1. contains the molecular formulae of only 2 potential biomarkers from 27. What were the selection criteria?

9) Sections 2.2.2 and 2.2.3 describe ROC analysis using all 27 biomarkers together, but not only two from section 2.2.1. I suppose that information about molecular formulae of all 27 potentially important molecules should be presented. Or the results of ROC analysis which demonstrate that only 2 biomarkers could be used to distinguish patients from controls should be demonstrated. However, all mentioned data could be interesting for readers.

Reviewer 2 Report

Comments and Suggestions for Authors

Dear authors,

Thank you for submitting your manuscript on "Assessment of Untargeted Metabolomics by Hydrophilic Interaction Liquid Chromatography – Mass Spectrometry to Define Breast Cancer Biomarkers in Liquid Biopsy" . The manuscript contains a large amount of information, and some promising results can be extracted.

However, I have some questions and comments.

1. Page 3/14, line 95. Please rewrite the sentence. The HILIC stationary phase can retain both hydrophilic (polar) and hydrophobic compounds (apolar), depending on how the injection conditions have been selected, which ion-pairing agents are used, what solvent the sample was dissolved in, how the column's equilibration has been performed, etc. 

2. In Figure 1, you show the exemplary chromatogram and state that these chromatograms show different peak profiles. I'm afraid I have to disagree with your statement. The chromatograms shown might differ in some details, but one cannot conclude from them. Would you please replace the current chromatograms with either extracted ion chromatograms for selected metabolites you have identified differently expressed in control and patients' samples or show a different chromatogram showing the difference more clearly?

3. Which features were "influenced by the MP"? (Page 3/14, line 11ff.) What substances were that, and how did you decide to exclude them and deem them the background?

4.  Page 6/14, line 156: Which molecular properties were selected to discriminate tentative candidates?

5. Page 7/14, line 200 ff. The mainly general description of possible analytes is insufficient. Please describe more precisely, which analytes you have defined as polar and which as non-polar. Currently, you state that  "may correspond...". However, they also may not correspond. Do you have a proof for this claim?

6. Page 8/14, line 216 ff. Please correct "death volume" to void volume! Please explain the poor retention of polar compounds and why they would elute nearly at t0.

7. Please explain the claim on page 8/14, line 218, "that most of the mass signals..." I don't understand what the information is here. 

8. Page 8/14, line 228 ff. Could you please rewrite or rephrase the sentence? What is meant by "acidic solutions", which acidic solutions? I understand you have added the reference, but the sentence is incomplete, and no information could be extracted.

9. The text on page 8/14 from 231 to the end of the paragraph is more suited for the introduction than the discussion because it does not discuss your results. 

Thank you and kind regards.

Comments on the Quality of English Language

Please check the grammar and style by enabling a native speaker to review the text. 

Round 2

Reviewer 1 Report

Comments and Suggestions for Authors

I would like to thank the authors for their thorough study of the comments and conduction of additional analyses. I suppose that the new version of the manuscript is much better and could be considered for the publication.

Reviewer 2 Report

Comments and Suggestions for Authors

Dear authors,

Thank you for taking the suggestions into account and making changes to the manuscript.

I have no further comments.

Kind regards.